# Development and Validation of a Reliable UHPLC-MS/MS Method for Simultaneous Quantification of Macrocyclic Lactones in Bovine Plasma

**DOI:** 10.3390/molecules27030998

**Published:** 2022-02-01

**Authors:** Gemechu Zeleke, Siegrid De Baere, Sultan Suleman, Mathias Devreese

**Affiliations:** 1Laboratory of Pharmacology and Toxicology, Department of Pathobiology, Pharmacology and Zoological Medicine, Faculty of Veterinary Medicine, Ghent University, B-9820 Merelbeke, Belgium; Gemechu.Carii@UGent.be (G.Z.); Siegrid.DeBaere@UGent.be (S.D.B.); 2Institute of Health, School of Pharmacy, Jimma University, Jimma P.O. Box 378, Ethiopia; sultan.sulemanl@gmail.com

**Keywords:** bovine, plasma, method development, macrocyclic lactones, UHPLC-MS/MS, bioanalysis

## Abstract

A fast, accurate and reliable ultra-high performance liquid chromatography–tandem mass spectrometry (UHPLC-MS/MS) method was developed for simultaneous quantification of ivermectin (IVER), doramectin (DORA), and moxidectin (MOXI) in bovine plasma. A priority for sample preparation was the eradication of possible infectious diseases to avoid travel restrictions. The sample preparation was based on protein precipitation using 1% formic acid in acetonitrile, followed by Ostro^®^ 96-well plate pass-through sample clean-up. The simple and straightforward procedure, along with the short analysis time, makes the current method unique and suitable for a large set of sample analyses per day for PK studies. Chromatographic separation was performed using an Acquity UPLC HSS-T3 column, with 0.01% acetic acid in water and methanol, on an Acquity H-Class ultra-high performance liquid chromatograph (UHPLC) system. The MS/MS instrument was a Xevo TQ-S^®^ mass spectrometer, operating in the positive electrospray ionization mode and two multiple reaction monitoring (MRM) transitions were monitored per component. The MRM transitions of *m*/*z* 897.50 > 753.4 for IVER, *m*/*z* 921.70 > 777.40 for DORA and *m*/*z* 640.40 > 123.10 for MOXI were used for quantification. The method validation was performed using matrix-matched calibration curves in a concentration range of 1 to 500 ng/mL. Calibration curves fitted a quadratic regression model with 1/x2 weighting (r ≥ 0.998 and GoF ≤ 4.85%). Limits of quantification (LOQ) values of 1 ng/mL were obtained for all the analytes, while the limits of detection (LOD) were 0.02 ng/mL for IVER, 0.03 ng/mL for DORA, and 0.58 ng/mL for MOXI. The results of within-day (RSD < 6.50%) and between-day (RSD < 8.10%) precision and accuracies fell within acceptance ranges. No carry-over and no peak were detected in the UHPLC-MS/MS chromatogram of blank samples showing good specificity of the method. The applicability of the developed method was proved by an analysis of the field PK samples.

## 1. Introduction

Malaria, a vector-borne disease caused by the *Plasmodium* parasite, continues to have a devastating impact on people’s health and lives around the world [1]. Malaria cases and deaths remain high in Africa, especially in children under the age of five [2]. The current increased insecticide resistance and zoophilic behavior in *Anopheles arabiensis* necessitate looking for new complementary vector control tools [3,4,5,6]. Zoo-prophylaxis of bovines with Macrocyclic Lactones (MLs) against *An. Arabiensis*, as it preferentially feeds on cattle, is a novel and complementary approach for malaria control and elimination, especially in East and Central Africa [7,8,9,10,11].

The avermectins (ivermectin (IVER) and doramectin (DORA)) and milbemycins (moxidectin (MOXI)) are 6-membered MLs endectocides commonly used against veterinary helminths, ectoparasites. They are semi-synthetic fermentation products of soil-dwelling bacteria of the genus Streptomyces with an excellent safety profile [12]. IVER is the prototype, and the commonly used endectocide, which is a mixture of 22, 23-dihydroavermectin B1a (>90%) and 22, 23-dihydroavermectin B1b (<10%). DORA (25-cyclohexyl-5-O-demethyl-25-de(1-methylpropyl)-avermectin A1a.) and MOXI (semi-synthetic methoxime derivative of nemadectin) are the newer generation MLs that have a potent action. MLs act by binding to glutamate-gated chloride channels in nerve and muscle cells of invertebrates, and cause paralysis in the neuromuscular system [13]. Currently, the use of MLs has been extended to humans, especially IVER, against onchocerciasis (river blindness) and is a candidate drug for malaria vector control when given to cattle targeting *An. Arabiensis* [14]. Pharmacokinetic (PK) data is available for the major breeds, including Holstein-Friesian [15]. However, PK data are lacking in cattle breeds in endemic regions, particularly in Ethiopia. To that aim, a fast, economic, and sensitive bioanalytical method is mandatory to allow accurate quantification of MLs in bovines. Furthermore, restrictions usually apply to ship samples from endemic regions to other countries based on the possible transmission of infectious agents, such as foot-and-mouth disease (FMD).

In an extensive review described by Danaher et al., 2006 [16], high-performance liquid chromatography-fluorescence (LC-FL) and high-performance liquid chromatography interfaced to tandem mass spectrometry (LC-MS/MS) have been utilized for the quantification of MLs in biological matrices. In LC-FL methods, MLs bioanalysis was only possible just after sample derivatization to produce fluorescent derivatives and requires a large matrix volume per analysis. The sample derivatization prior to LC-FL bioanalysis is more laborious, time-consuming, and not suitable in the case of large sets of PK sample analysis. On top of this, the rapid degradation of the MLs just after derivatization, as described by de Montigny et al., 1990 [17], Cerkvenik, 2001 [18], Danaher et al., 2006 [16], and Kolberg et al., 2009 [19], makes LC-FL methods less suitable, and consequently, majorly limits its use for PK studies unless online derivatization is devised. In fact, this may also be dependent on the protocol used during sample derivatization. 

However, high-performance liquid chromatography interfaced to tandem mass spectrometry (LC-MS/MS) has exhibited multiple advantages, facilitating fast, more sensitive, and accurate analysis of drugs and metabolites in biological matrices. LC-MS/MS also allows accurate simultaneous analysis of mixtures in biological matrices, such as plasma. A few studies have been reported for quantification of MLs using LC-MS/MS methods in biological matrices, such as in milk [20], lamb tissues [21,22], and lamb serum [23], human [24], dog [25], and calf [26] plasma.

To date, LC-MS/MS analysis of MLs in cattle plasma has only been described by Croubels et al., 2002. Even though this method has been successfully applied for IVER, it demands a large volume matrix and further sample clean-up to apply the method to the other MLs [26]. As a result, a novel sample preparation technique is mandatory for the fast and accurate quantification of MLs in bovine plasma. One of the new approaches is the use of an Oasis Ostro^®^ Protein Precipitation & Phospholipid Removal 96-well plate, which combines protein precipitation using an organic solvent with the removal of phospholipids to reduce the matrix effects via a pass-through principle. Due to its simplicity, reproducibility, and highly efficient phospholipid removal, the Oasis Ostro^TM^ 96-well plate pass-through technique is superior and attractive for obtaining high-quality data in LC-MS/MS analysis [27,28,29]. 

The objective of this study, therefore, was to develop, optimize and validate a method for fast, accurate, and reliable simultaneous quantification of the MLs (IVER, DORA, and MOXI) in bovine plasma using UHPLC-MS/MS, deploying a straightforward sample clean-up procedure for its application in PK-PD studies. Special attention was given to a sample preparation that inactivates infectious agents that would otherwise hamper sample shipment to several countries.

## 2. Results and Discussion

### 2.1. Method Development

The simplicity and suitability of the procedure for large sample set analysis were taken into consideration during the UHPLC-MS/MS method development and optimization. The aim was to develop a very short, simple, and inexpensive sample preparation procedure to allow the extraction of ≥96 samples in one batch. Moreover, a UHPLC-MS/MS method with a short run time (~12 min) was developed to allow the analysis of the MLs in the large sets of plasma samples (*n* ≥ 96) within a day. Finally, the developed UHPLC-MS/MS method was fully validated in-house to verify its specificity, reliability, and sensitivity for quantification of the MLs (IVER, DORA, and MOXI) in bovine plasma.

#### 2.1.1. Optimization of Sample Preparation and Clean-Up

The sample preparation was initiated based on previous studies described by Croubels et al., 2002 (IVER in calf plasma) [26] and Morbidelli et al., 2018 (IVER in dog plasma) [25]. The protein precipitation solvents acetonitrile, methanol, and ethanol were evaluated and were added to the plasma samples in a ratio of 75/25 (*v*/*v*). Even though protein precipitation using acetonitrile showed good extraction recovery, a fluctuating and low signal intensity was observed for MOXI and DORA, resulting in poor method accuracy and precision (results not shown). This signified the need for a further clean-up of the plasma extract using solid-phase extraction (SPE) to remove fats, phospholipids, and other co-eluting components.

Two SPE sorbents using a simple and fast pass-through protocol were evaluated during preliminary experiments, i.e., Oasis^®^ PriME HLB and OstroTM (both from Waters). Veterinary drugs pass through the Oasis^®^ RriME HLB column, while the sorbent holds back interferences [30]. The OstroTM sorbent removes proteins, particulates, and more than 95% of phospholipids from the sample matrix. The Ostro^TM^ 96-well plate utilizes protein precipitation solvent in combination with a single, rapid, pass-through method [29]. Three protein precipitation solvents were tested for the extraction of standard solutions (100 ng/mL) i.e., 1% formic acid (FA) in acetonitrile, methanol, and ethanol. As can be seen in Appendix A-Appendix A, good extraction recovery was obtained while using Ostro^TM^ 96-well plate (IVER, DORA, and MOXI were 84.8, 97.0, and 98.1%, respectively) relative to the Oasis^®^ PriME 96-well µ-Elution plate. Moreover, 1% FA in € was the good deprotinisation solvent along with OstroTM 96-well plate pass-through SPE and was therefore selected in the final procedure. A schematical overview of the final sample preparation procedure for quantification of IVER, DORA, and MOXI in bovine plasma is indicated here (see Figure 1).

Compared to other methods in the literature, the current method is fast and simple to apply since drying and reconstitution steps were absent with less solvent consumption; moreover, the procedure is rather inexpensive, since OstroTM 96-well plates are substantially cheaper than conventional SPE columns [23,25] and samples do not need to be filtrated using 0.22 µm filters or transferred into autosampler vials, and hence, can be directly injected from the 96-well collector plate onto the LC-MS/MS instrument. As a result, the developed sample preparation procedure is suitable for large sample set extraction within a short period of time. Moreover, OstroTM 96-well plate was effectively applied for the analysis of other veterinary drugs and mycotoxins in plasma samples of different animal species, such as chicken, turkey, cattle, and pig (Lauwers et al., 2019) [31], De Baere et al., 2018 [32], De Baere et al., 2015 [33].

Finally, the acidification using formic acid was essential to reduce the pH < 6, effectively inactivating viral agents such as FMD. This has a significant impact on the shipping restrictions of the samples.

#### 2.1.2. Internal Standardization

Deuterated internal standards that display similar properties to the analytes, were used to enhance method performance and reduce analytical variations, majorly resulting from analyte loss during sample preparation and from matrix effects due to the co-eluting endogenous components during UHPLC-MS/MS analysis. Stable isotopically labeled internal standards, which could only be commercially obtained at a reasonable price for IVER and MOXI (i.e., ivermectin-d2 and moxidectin-d3), were used. These isotopically labeled internal standards have the same molecular and physicochemical properties as the analytes of interest and elute at the same retention time [34,35].

Ivermectin-d2 (for IVER and DORA) and moxidectin-d3 (for MOXI) were used with success as internal standards, as can be seen from the validation results (see Section 2.2).

#### 2.1.3. Optimization of Chromatographic Conditions

Chromatographic conditions were optimized to develop a simple and reliable method for simultaneous quantification of IVER, DORA, and MOXI in bovine plasma within a total run time of not more than 12 min. Two reverse phase UPLC columns (Acquity BEH C18 column (50 × 2.1 mm i.d., dp: 1.8 μm) and an Acquity HSS-T3 column ((100 × 2.1 mm), dp: 1.8 μm), both from Waters, were evaluated for the chromatographic separation of the analytes in plasma. The Acquity HSS T3 100 × 2.1 mm UPLC column (1.8 μm particle size) in combination with an Acquity HSS T3 1.8 μm Vanguard pre-column showed an optimal peak separation with good signal intensity and peak shape compared to the Acquity BEH C18 column and was therefore selected for further experiments (results not shown). Similarly, in a previous study, an Acquity HSS T3 100 × 2.1 mm UPLC column (1.8 μm particle size) in combination with an Acquity HSS T3 1.8 μm Vanguard pre-column was also used for the multi-residue analysis of veterinary drugs, including ivermectin in pond water as described by Goessens et al., 2020 [36].

A mixture of ULC/MS grade aqueous and organic solvents (acetonitrile and methanol) in combination with organic modifiers (formic acid (FA) and acetic acid (AA)) were evaluated in several compositions and gradient programs to evaluate the chromatographic baseline separation, signal intensity, peak shape, and retention time. The addition of 0.1% AA to both the aqueous and organic phase (acetonitrile) resulted in a better signal for DORA, compared to water and 0.1% FA, whereas for IVER the best results were obtained by using no organic modifier (see Appendix A). MOXI was not tested during the initial experiment, but Chhonker et al. 2018 [24] also previously reported 0.1%AA as an organic modifier for MOXI. By reducing the AA concentration from 0.1% to 0.01%, the signal intensity for IVER and MOXI increased, whereas a decreased signal intensity for DORA was observed (see Appendix A). As a compromise result, 0.01% acetic acid was selected as the final concentration of the organic modifier since this was also successfully applied by Croubels et al., 2002 [26]. Next, the influence of the organic phase on signal intensity was evaluated. As can be seen in Appendix A, good signal intensity was obtained for all components while 0.01% AA in methanol was used as an organic phase, relative to 0.01% AA in acetonitrile. 

Moreover, the retention time of the analytes increased using methanol as an organic phase, compared to acetonitrile (see Appendix A). The mobile phase gradient was further optimized to achieve a total run time of a maximum 12 min, which is much shorter than previous methods reported by Inoue et al., 2009 [22], analysis of tissues (30 min); Moshou et al., 2019 [37], analysis of fish tissue (20 min); however, comparable with the methods reported by Morbidelli et al., 2018 [25], analysis in milk (11 min); Croubels et al., 2002 [26], analysis of calf plasma (6 min).

#### 2.1.4. Optimization of MS/MS and Multiple Reaction Monitoring (MRM) Parameters

Optimization of the MS/MS parameters was performed by infusion of 1 µg/mL standard solutions of each analyte and IS’s into the Xevo TQ-S^®^ mass spectrometer at a flow rate of 10 µL/min, in combination with the mobile phase (20% A/80% B, flow rate: 0.2 mL/min) using the IntelliStart Fluidics system.

Good sensitivity for all analytes and their respective ISs was obtained when the mass spectrometer was operated in the positive electrospray ionization (ESI) mode. While operating in the full scan MS mode, the mass spectra for IVER, IVER-d2, and DORA showed the sodium adduct ions [M + Na]+ as the base peak, which were selected as the precursor ions (at *m*/*z* 897.50, 899.50, and 921.70, respectively). For MOXI and MOXI-d3, the [M + H]+ ion was chosen as the precursor ion (at *m*/*z* 640.40 and 643.50, respectively) having similarity with Baptista et al., 2017 [23], Michele et al., 2018 [22], and Hofmann et al., 2021 [38]. The precursor ion of each analyte was fragmented and monitored for product ions at different collision energies and cone voltages. For each precursor ion, the two most abundant product ions were monitored and used for quantification and identification purposes, respectively (see Table 1). The product ions for IVER, DORA, and MOXI were in accordance with other reports in the literature (Croubels et al., 2002 [26], Wang et al., 2011 [39], and Li et al., 2017 [40]). For MOX, the most intense product ion with *m*/*z* = 622 corresponds with the [M-H2O + H]+. However, the product ion with *m*/*z* = 123 was used as a quantifier ion since method performance was better compared to the use of the product ion with *m*/*z* = 622 as quantifier ion (results not shown).

### 2.2. Method Validation

In this study, the method validation parameters including linearity, the limit of detection (LOD), limit of quantification (LOQ), precision, accuracy, carry-over, specificity, and stability were evaluated to determine the UHPLC-MS/MS method performance. The current UHPLC-MS/MS method with a straightforward sample preparation (protein precipitation with 750-µL 1% formic acid in acetonitrile followed by Ostro^®^ 96-well plate pass-through clean-up), met all requirements. 

#### 2.2.1. Linearity

The linearity of the method was evaluated by the correlation coefficie€(r), goodness-of-fit coefficients (GoF), and the back-calculated concentrations. During the validation, matrix-matched calibration curves were freshly prepared on 3 different analysis days from blank cattle plasma that was spiked with the analytes of interest at 10 concentrations (1, 2.5, 5, 10, 25, 50, 75, 100, 250, and 500 ng/mL). The selected concentration range of the calibration curve was based on the concentrations that were determined in incurred bovine plasma samples.

Calibration curves were best constructed using weighted (1/x2) quadratic regression analysis (y = ax2 + bx + c, with y = peak area ratio analyte/IS for IVER, DORA, and MOXI). The correlation coeffi€nt (r) values ranged from 0.9985 to 0.9997 and goodness-of-fit coefficients (GoF) were between 1.91 to 4.85% (see Table 2), which complied with the acceptance criteria of r > 0.99 and GoF < 10% [41]. The back-calculated concentrations of the analytes in each calibrator sample fulfilled the criterion for accuracy (results not shown). Calibration curves were broader than those reported elsewhere by Croubels et al., 2002 [26], Baptista et al., 2017 [23], and Morbidelli et al., 2018 [25].

#### 2.2.2. LOQ and LOD

As can be seen in Table 2, LOQ values of 1 ng/mL were reached for IVER, DORA, and MOXI which was the same as the previous report by Croubels et al., 2002 [26] and was more sensitive than the previous studies reported in lamb tissues [21,22] and lamb serum22 [23]. This LOQ was low enough to allow quantification of the MLs after 4 h and until 35 days after a single subcutaneous administration of 0.2 mg/kg BW of IVER, DORA, and MOXI to cattle. The LOD was calculated based on the signal-to-noise ratio (S/N) of the analyte peak in the LOQ samples. The theoretical concentration that corresponded with an S/N ratio of 3/1 was set as the LOD. In this study, the calculated LOD values were 0.02 ng/mL, 0.03 ng/mL and 0.58 ng/mL for IVER, DORA, and MOXI, respectively [41].

#### 2.2.3. Precision and Accuracy

Within-day precision and accuracies were evaluated at the limit of quantification level (1 ng/mL) and at a low (5 ng/mL), middle (50 ng/mL), and high (250 ng/mL) concentration level (in 6 replicates each). Similarly, between-day precision and accuracy were evaluated by combining the results of the three data sets obtained for within-run precision and accuracy. The results of within-day and between-day precision (RSD, %) ranged between 1.1% to 6.50 and 2.3% to 8.10%, respectively, which comply with VICH GL49 acceptance criteria, whereas accuracies all fell within the acceptance limit at the specified concentration levels [41,42] (see Table 3). From the one-way ANOVA analysis using STATA software, there was no statistically significant difference between means of accuracies (*p* = 0.1480) and precisions (*p* = 0.1594) for each of the drug groups during the three different days of the validation experiment. In pairwise comparison, no significant differences were also found for MOXI (*p* > 0.315), DORA (*p* > 0.712), and IVER (*p* = 1.000) with respect to accuracy and precision (*p* = 1.000) showing the repeatability, reproducibility, and accurateness of the analytical method.

#### 2.2.4. Carry-Over and Specificity

The carry-over on the LC-MS/MS instrument was evaluated by the injection of a solvent sample after the highest calibrator sample. No carry-over was observed at or near the retention time of each of the analytes and respective internal standards (see Figure 2B).

With regard to specificity, no peaks were observed at the elution zone of each analyte of interest and respective internal standards, demonstrating the specificity of the developed method for the analysis of IVER, DORA, and MOXI, as can be seen in Figure 2A.

#### 2.2.5. Matrix Effect (ME) and Extraction Recovery (RE)

The efficiency to precipitate proteins and extract the analytes of interest from the plasma matrix (RE) and the signal enhancement or suppression due to matrix effect (ME) of the sample extracts on the LC-MS/MS instrument were evaluated for the analytes of interest IVER, DORA, and MOXI, using the final procedure, and the results are shown in Figure 3. As can be seen, RE ranged between 37.0 and 52.8% (see Figure 3A), whereas the ME was between 105.2–123.7% (see Figure 3B). Despite the low extraction recovery (RE), the isotopically labeled internal standards (IVER-d2 and MOXI-d3) effectively compensated the analyte losses in all three drugs (IVER, DORA, and MOXI) and maintained reproducibility and accuracy of the method. In this final method, the matrix effect (signal suppression or enhancement) due to co-eluting undetected matrix components was sufficiently minimized by using the OstroTM 96-well plates clean-up procedure. Furthermore, matrix-matched calibrator samples and the isotope-labeled internal standards used in the present method have also contributed to this minimum matrix effect. With respect to matrix effects, the coefficient of variation (CV) for IVER, DORA, and MOXI was 13%, 5%, and 4%, respectively, which was in line with the acceptance limit VICH GL49 [41] and EU recommendation [42]. 

#### 2.2.6. Stability

All the analytes were stable in stock and working solutions during storage for at least 21 months (results not shown). As can be seen in Table 4, all the analytes were found to be stable in the sample extract and during three freeze-thaw cycles (≤−15 °C to room temperature). Sample extracts were stored in capped 96-well collector plates for 12 days at 2–8 °C, which corresponds with the autosampler temperature of the LC-MS/MS instrument. The mean found concentration fell within the acceptance ranges for accuracy and precision for all analytes at the tested concentration levels (5 and 50 ng/mL, see Table 4). These results demonstrate that analyte concentrations were not significantly affected by the applied extraction method, thus showing the practicability of the method for quantification of the analytes in bovine plasma.

The long-term stability of the analytes of interest was studied in acidified plasma (containing 0.5% FA, pH~4), since it was the aim to ship these acidified plasma samples from Ethiopia to Belgium, where the final sample preparation and UHPLC-MS/MS analysis were performed.

The blank bovine plasma that was acidified with 0.5% formic acid and spiked at an analyte concentration level of 100 ng/mL was extracted and analyzed after a storage period of 12 and 28 days and 9 months at ≤−15 °C (*n* = 3 per time point). At each time point, mean peak area ratios (analyte/IS) were determined and compared with peak area ratios at the time of preparation (day 0). As can be seen from Figure 4, the calculated % recoveries fell within 90 ± 10% for all analytes, indicating that no significant degradation occurred within a storage period of 28 days. Over a long-term storage period (9 months) at ≤−15 °C, analyte concentrations decreased between 15–20% of the initial value, which is still acceptable.

#### 2.2.7. Analysis of Biological Samples

The applicability of the current method was proved by UHPLC-MS/MS analysis of field PK samples, along with quality control samples using a matrix-matched calibration curve (concentration range from 1–500 ng/mL). As can be seen in Figure 5, analyte concentrations above the LOQ level (1 ng/mL) could be detected for all analytes until 21 days after a single subcutaneous administration of 0.2 mg/kg BW of IVER, DORA, and MOXI, showing the applicability of the developed method for future PK studies with these components in bovine animal species. This section may be divided into subheadings. It should provide a concise and precise description of the experimental results, their interpretation, as well as the experimental conclusions that can be drawn.

## 3. Materials and Methods

### 3.1. Chemicals, Products and Reagents 

All the standards (ivermectin (IVER), doramectin (DORA), moxidectin (MOXI), ivermectin-d2 (IVER-d2), and moxidectin-d3 (MOXI-d3)) were obtained from Sigma-Aldrich (Bornem, Belgium). All the standards were stored at ≤−15 °C. UPLC-MS grade acetonitrile, methanol, acetic acid, and formic acid obtained from Biosolve BV (Valkenswaard, The Netherlands) were used for the preparation of mobile phases. While, the ultrapure water used in the mobile phase was from a Milli-Q-SP reagent water system (Merck Millipore, Overijse, Belgium). Oasis Ostro^®^ Protein Precipitation & Phospholipid Removal 96-well plates used for sample clean-up were purchased from Waters (Zellik, Belgium).

### 3.2. Blank Plasma Samples 

Blank cattle plasma used for matrix-matched calibrator and quality control samples was obtained from donor cattle maintained at the Department of Large Animal Internal Medicine, Faculty of Veterinary Medicine, Ghent University. The animal was healthy and had no past history of drug therapy with macrocyclic lactones. The blank plasma was acidified with formic acid to achieve a final 0.5% formic acid in blank bovine plasma with a pH < 6. The 0.5% formic acid in blank bovine plasma was prepared by the addition of 10% formic acid in a water solution into a 15-mL tube containing 10 mL blank plasma, followed by vortex mixing. 

### 3.3. Incurred Plasma Samples 

For a demonstration of the applicability of the UHPLC-MS/MS method, plasma samples from a pharmacokinetic (PK) study with MLs in bovines were analyzed. The drugs IVER, DORA, and MOXI were administered to local Ethiopian bovines (Bos indicus) once subcutaneously (dose: 0.2 mg/kg body weight) in a parallel study which was performed in Ethiopia after ethical review and approval (Ref. No. IUC-JU/M45/12) by the Institutional Animal Care and Use Committee at the veterinary facility of Jimma University, Ethiopia. Blood samples were collected from each animal before and periodically post-administration (p.a.), i.e., at 4, 8, 12, 24, and 48 h, and at 3, 5, 7, 10, 14, 21, 28, and 35 days, using EDTA vacutainer tubes. About 5 mL of a blood sample was collected from the left jugular vein. The collected blood samples were immediately put in an ice box and then centrifuged (within two hours) for 20 min at a maximum of 3000 rpm. The plasma was transferred into plastic tubes and stored in a deep freezer at ≤−15 °C. 

To inactivate animal-borne viruses, plasma samples were treated with formic acid (0.5%) to lower the pH < 6. Therefore, 1000 µL of plasma was transferred into a tube followed by the addition of 50 µL of a 10% FA solution in water. After vortex mixing, the samples were stored again in the freezer at ≤−15 °C until shipment to Belgium (within two weeks).

### 3.4. Preparation of Standard Stock and Working Solutions

The standard stock solutions were prepared using acetonitrile as a solvent. IVER, DORA, MOXI, IVER-d2, and MOXI-d3 stock solutions were prepared at a concentration of 1 mg/mL based on the percent purity information from the manufacturer and stored at ≤−15 °C. The stock solutions of all analytes were stable for at least one year under these storage conditions.

Mixed working solution (WS_mix_) of IVER, DORA, and MOXI (10 µg/mL) in acetonitrile was prepared by transferring 100 µL of each 1 mg/mL stock solution in a volumetric flask of 10.0 mL and addition of acetonitrile up to the mark, followed by gently mixing and equilibration for 5 min at room temperature. Further, WS_mix_ solutions with concentrations of 5000, 2500, 1000, 750, 500, and 250 ng/mL were prepared by appropriate dilution of WS_mix_ 10 µg/mL with acetonitrile. Similarly, the WS_mix_ solutions with concentrations of 100, 50, 25, and 10 ng/mL were prepared from the WS_mix_ 1000 ng/mL by appropriate dilution with acetonitrile.

Individual working solutions (WS_ind_) of 100 µg/mL ivermectin-d2 and moxidectin-d3 were prepared separately by diluting 100 µL of 1 mg/mL stock solution with 900 µL acetonitrile in an Eppendorf cup and vortex mixed. Further, a mixed working solution containing 1 µg/mL moxidectin-d3/ ivermectin-d2 (WS_IS_mix_) was prepared by transferring 100 µL of each WS_ind_ 100 µg/mL of moxidectin-d3 and WS_ind_ 100 µg/mL of ivermectin-d2 in a volumetric flask of 10.0 mL and with an addition of acetonitrile up to the mark, followed by gently mixing. All working solutions were stored at ≤−15 °C.

### 3.5. Sample Preparation

The sample preparation was based on a deproteinisation with 750-µL of a 1% formic acid in acetonitrile solution, followed by an Ostro^®^ 96-well plate pass-through clean-up. 

For deproteinisation, 250-µL of the plasma sample was transferred to an Eppendorf cup, followed by the addition of 25-µL of the WS_IS_mix_ (1 µg/mL) and vortex mixing for 15 s. For the preparation of calibration/quality control samples with analyte concentrations of 1, 2.5, 5, 10, 25, 50, 75, 100, 250, and 500 ng/mL, and 225 µL of blank plasma was transferred to an Eppendorf cup and spiked with 25 µL of WS_mix_ 10, 25, 50, 100, 250, 500, 750, 1000, 2500 and 5000 ng/mL, respectively. Then, 25 µL of the WS_IS_ mix_ (1 µg/mL) was added to each calibrator and quality control sample, followed by vortex mixing for 15 s and equilibration for 5 min at room temperature.

To each of the above (spiked) samples (including the blank sample), 750 µL of a 1% formic acid in acetonitrile solution were added and vortex mixed for 5 min on a multi-tube vortex mixer (2500 rpm, BenchMixer^TM^ XLQ, Sigma-Aldrich, Overijse, Belgium). Further, the samples were centrifuged at 13,000 rpm for 10 min. at 4 °C using a microcentrifuge (Biofuge Fresco, Sysmex, Hoeilaart, Belgium). 

For further sample clean-up, the supernatant of each of the above samples was carefully transferred to an Ostro™ 96-well plate and closed by a polypropylene cap-mat. Then, vacuum was applied for 5 min allowing the transfer of each sample to a 2 mL square 96-well collector plate. The polypropylene cap-mat was replaced by a pre-slitted silicone/PTFE treated cap-mat before transfer to the autosampler of the UPLC-MS/MS instrument. Finally, a 10-µL aliquot was injected onto the UPLC-MS/MS instrument for analysis.

### 3.6. UHPLC-MS/MS Instrumentation 

An Acquity H-Class ultra-high performance liquid chromatograph (UHPLC) system consisting of an Acquity UPLC H-Class Quaternary Solvent Manager and Flow-Through-Needle Sample Manager with temperature-controlled tray and column oven from Waters (Zellik, Belgium) was used. Chromatographic separation was achieved on an Acquity UPLC HSS T3 column (100 mm × 2.1 mm i.d., dp: 1.8 µm) in combination with an Acquity HSS T3 1.8 μm Vanguard pre-column, both from Waters.

Mobile phase A consisted of 0.01% acetic acid in water, while mobile phase B was 0.01% AA in methanol. A gradient elution was performed: 0–0.5 min (20% A, 80% B), 6.0 min (linear gradient to 99% B), 6.0–7.7 min (1% A, 99% B), 8.0 min (linear gradient to 20% A), 8.0–12.0 min (20% A, 80% B). The flow rate was 0.3 mL/min. The temperatures of the column oven and autosampler tray were set at 40 °C and 8 °C, respectively. 

The UPLC column effluent was interfaced to a Xevo® TQ-S triple quadrupole tandem mass spectrometer system (MS/MS), equipped with an electrospray ionization (ESI) probe operating in the positive mode. A divert valve was used and the UPLC effluent was directed to the mass spectrometer from 5.5 to 8.0 min to avoid contamination and pollution of the mass spectrometer and to maintain sensitivity for a longer period.

Instrument parameters were optimized by direct infusion of working solutions of 100 ng/mL of all analytes and the ISs at a flow rate of 10 µL/min and in combination with the mobile phase (20% A, 80% B, flow rate: 200 µL/min). 

The following Xevo^®^ TQ-S mass spectrometer settings were used: capillary voltage: 3.5 kV; source offset: 60 V; source temperature: 150 °C; desolvation temperature: 500 °C; desolvation gas: 800 L/h; cone gas: 150 L/h; nebuliser pressure: 6.9 bar; LM resolution 1 and 2: 2.8; HM resolution 1 and 2: 15; ion energy 1 and 2: 0.2 and 0.8, respectively; collision gas flow: 0.15 mL/min. 

MS/MS acquisition was performed, simultaneously for all the MLs in the plasma sample, in the multiple reaction monitoring (MRM) mode. The MRM transitions that were monitored are shown in Table 1. Data acquisition and processing were performed using the MassLynx^®^ and TargetLynx^®^ software (all from Waters).

### 3.7. Determination of Extraction Recovery (RE) and Matrix Effects (ME)

Extraction recovery and matrix effects were evaluated based on the post-extraction spike method described by Matuszewski et al., 2003 [43] and Chambers et al., 2007 [44]. Nine cattle plasma samples were prepared in three series’ (A-series, B-series, and C-series) using three different biological samples in each series. The C-series samples were neat standard solutions prepared by spiking 250 µL of water with 25 µL of WS_mix_ 25 ng/mL and 25 µL of WS_IS_mix_ 1 µg/mL, followed by the addition of 750 µL of 1% FA in acetonitrile.

The B-series samples consisted of blank plasma that was subjected to the sample preparation procedure described above. Just after passing the supernatant of blank extract through the Oasis Ostro^®^ 96-well plate, 25 µL of WS_mix_ 25 ng/mL and 25 µL of WS_IS_mix_ 1 µg/mL were added to the sample extract in the collector plate.

The A-series samples consisted of blank plasma that was spiked prior to the sample preparation with 25 µL of WS_mix_ 25 ng/mL and 25 µL of WS_IS_mix_ 1 µg/mL. After vortex mixing, the samples were subjected to the sample preparation procedure, as described above in Section 3.5.

The RE and ME were determined based on the absolute peak areas of the analytes in the different samples using the following formulas: RE (%) = (A-series peak area/B-series peak area) × 100(1)
ME (%) = (B-series peak area/C-series peak area) × 100(2)

### 3.8. Evaluation of Stability of IVER, DORA and MOXI in Acidified Plasma 

To allow the import of plasma samples from Ethiopia to Belgium, the pH of the plasma samples had to be <6. Therefore to assess the stability of the analytes in acidic pH, formic acid was added to blank plasma at different percentages (3%, 1%, 0.5%, 0.1%, and 0%) and the pH was recorded. In addition, the short-term stability of IVER, DORA, and MOXI in the acidified plasma was evaluated.

To 2700 µL of (acidified) blank plasma (% FA ranging from 0% to 3%), 300 µL of WS_mix_ 1 µg/mL was added, followed by vortex mixing for 15 sec and equilibration for 5 min at room temperature (final concentration of IVER, DORA, and MOXI: 100 ng/mL). The spiked (acidified) plasma was divided into 250-µL aliquots into Eppendorf cups and stored at <−15 °C until the moment of sample preparation and UHPLC-MS/MS analysis. Three 250-µL aliquots of each spiked (acidified) plasma were analyzed the same day of preparation (day 0). The remaining aliquots were thawed, extracted, and analyzed at day 12, day 28 and after 9 months, along with a freshly prepared spiked plasma sample (25 µL WS_mix_ 1.0 µg/mL added to 225 µL blank plasma, vortex mixed and equilibrated for 5 min at room temperature). The peak area ratios (analyte versus internal standard) in the spiked (acidified) plasma after storage at ≤−15 °C (test) were compared with the corresponding peak area ratios in freshly spiked plasma samples that were not acidified with formic acid (reference). The percentage analyte recovery was calculated as follows: % recovery = (peak area ratio test/peak area ratio reference) × 100(3)

### 3.9. Method Validation 

The UHPLC-MS/MS method developed for the quantification of IVER, DORA, and MOXI in bovine plasma was validated based on international guidelines [41,42,45]. The following parameters were evaluated: linearity, within-day precision, and accuracy, between-day precision and accuracy [41], the limit of quantification (LOQ), limit of detection (LOD), carry-over, specificity, and stability. All the parameters were evaluated using spiked blank acidified (containing 0.5% formic acid) bovine plasma obtained from a healthy, untreated animal.

#### 3.9.1. Linearity

Matrix-matched calibrator samples with IVER, DORA, and MOXI concentrations of 1, 2.5, 5, 10, 25, 50, 75, 100, 250, and 500 ng/mL were prepared by spiking 225 µL of blank acidified plasma with 25-µL of WS_mix_ 10, 25, 50, 100, 250, 500, 750, 1000, 2500 and 5000 ng/mL, respectively. Calibration curves were prepared freshly on three consecutive days. The correlation coefficient (r) and goodness-of-fit coefficient (GoF) % were evaluated and had to be ≥0.99 and ≤20%, respectively. Moreover, the back-calculated concentration of the analytes in each calibrator sample had to fulfill the criterion for accuracy. Evaluation of the weighting factor was based on the sum of the GoF-factors of the 3 calibration curves after applying unweighted, 1/x, and 1/x2 weighted regression analysis. The weighting factor that resulted in the smallest sum of GoF was considered as most appropriate.

#### 3.9.2. Precision and Accuracy

Quality control (QC) samples with analyte concentrations at the limit of quantification level (1 ng/mL) and at a low (QC-L, 5 ng/mL), medium (QC-M, 50 ng/mL), and high (QC-H, 250 ng/mL) [41,42] concentration level were prepared to evaluate within-run precision and accuracy. Six replicates at each concentration level were prepared on three different analysis days. Similarly, between-day precision and accuracy were evaluated by combining the results of the three data sets obtained for within-run precision and accuracy. The acceptance criteria for within-day and between-day accuracy was considered being between −20% to +10%, −30% to +10% and −40% to +20% for analyte concentrations of ≥100 ng/mL, ≥10 to <100 ng/mL and ≥1 ng/mL to <10 ng/mL in cattle plasma, respectively, according to VICH GL49 [41] and EU recommendations [42]. Within-day and between-day precisions were evaluated based on the relative standard deviation (RSD%) and maximum standard deviation (RSDmax), respectively. The acceptance criteria for within-day precision were as follows: RSD (%) <25%, <15% and <10% for analyte concentrations ranging ≥1 to <10 ng/mL, ≥10 to <100 ng/mL and ≥100 ng/mL, respectively. For between-day precision, calculated RSD (%) values had to be lower than the RSDmax value as determined by the Horwitz equation with RSDmax = 2(1–0.5 log C) with C = concentration expressed as a decimal fraction (e.g., 1 ng/mL is entered as 10^−9^). Accordingly, the acceptance criteria for between-day precision were set at RSD (%) ≤32%, 23% and 16% for analyte concentrations ranging between ≥1 to <10 ng/mL, ≥10 to <100 ng/mL and ≥100 ng/mL, respectively [41,42]. 

#### 3.9.3. Limit of Quantification (LOQ) and Limit of Detection (LOD)

The LOQ was the smallest measured concentration of the analytes above which the accuracy and precision were acceptable. The LOD was the smallest measured concentration of an analyte from which it was possible to deduce the presence with acceptable certainty. The LOD was determined by calculating the theoretical analyte concentration that corresponded with a signal-to-noise (S/N) ratio of 3/1, based on the S/N ratio of the analytes in the LOQ samples [41,42]. 

#### 3.9.4. Carry-Over and Specificity

The carry-over was evaluated by injecting a solvent sample just after the highest calibrator sample. The specificity of the method was evaluated by analyzing a blank matrix sample extract. The response of the peak that eluted eventually at the same retention time as the analyte of interest should not be more than 20% of the mean peak area of the analytes in LOQ samples [41,42]. 

#### 3.9.5. Stability

Freeze/thaw stability during three cycles (≤−15 °C to room temperature), stability in sample extracts during storage at 2–8 °C for 12 days, and long-term stability in acidified plasma matrix (containing 0.5% FA) that was stored at ≤−15 °C for 12 and 28 days and 9 months was assessed using blank cattle plasma spiked with the analytes of interest at a low (5 ng/mL), medium (50 ng/mL) or high (100 ng/mL) concentration levels. For the freeze/thaw stability and stability in extract experiment, quantification was performed using freshly prepared matrix-matched calibration curves.

## 4. Conclusions

A sensitive and reliable UHPLC-MS/MS method for simultaneous quantification of IVER, DORA, and MOXI in bovine plasma was developed and validated. The sample preparation consisted of straightforward deproteinisation followed by Ostro^TM^ 96-well plate clean-up which was high-throughput and suitable for large number PK sample studies. With respect to all the validation parameters, the results of the current method fell within the set acceptance ranges.

The applicability of the validated method for field pharmacokinetic study was proved by the analysis of part of the ongoing pharmacokinetic samples. Results showed that the LOQ values (1 ng/mL) were sufficiently low and that the calibration range (1–500 ng/mL) was appropriate to allow a proper quantification of IVER, DORA, and MOXI. Moreover, due to the simple and straightforward sample preparation procedure in combination with a short analysis time, the method was suitable for the analysis of a large number of plasma samples (*n* ≥ 96) per day. This proved the importance of the current method for use in large PK studies.

## Figures and Tables

**Figure 1 molecules-27-00998-f001:**
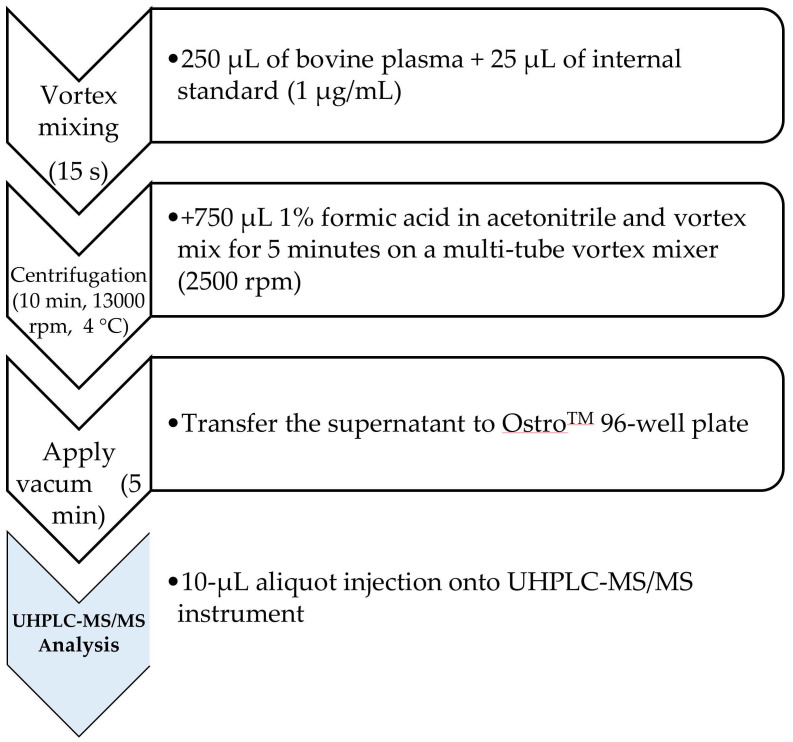
Schematical overview of the final sample preparation procedure for quantification of ivermectin (IVER), doramectin (DORA), moxidectin (MOXI) in bovine plasma.

**Figure 2 molecules-27-00998-f002:**
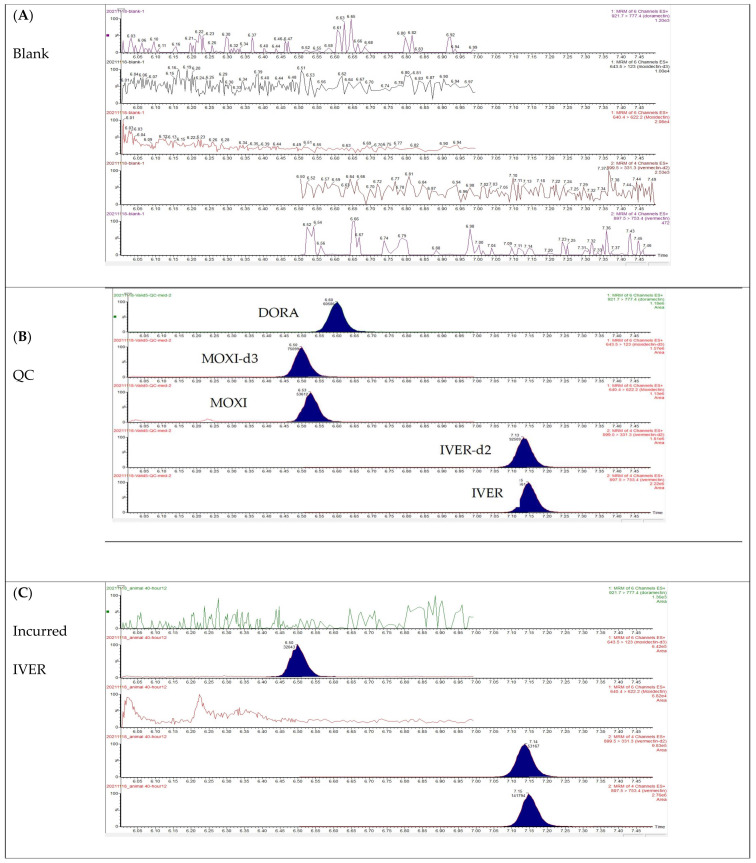
UHPLC-MS/MS chromatogram of (**A**) a blank plasma, (**B**) a quality control (QC) sample spiked with all analytes at a concentration of 50 ng/mL and incurred plasma samples containing (**C**) IVER (concentration: 128.89 ng/mL), (**D**) MOXI (concentration: 99.84 ng/mL), and (**E**) DORA (concentration: 14.60 ng/mL) that were extracted using 1% formic acid in acetonitrile as deproteinization solvent, followed by OstroTM 96-well plate clean-up.

**Figure 3 molecules-27-00998-f003:**
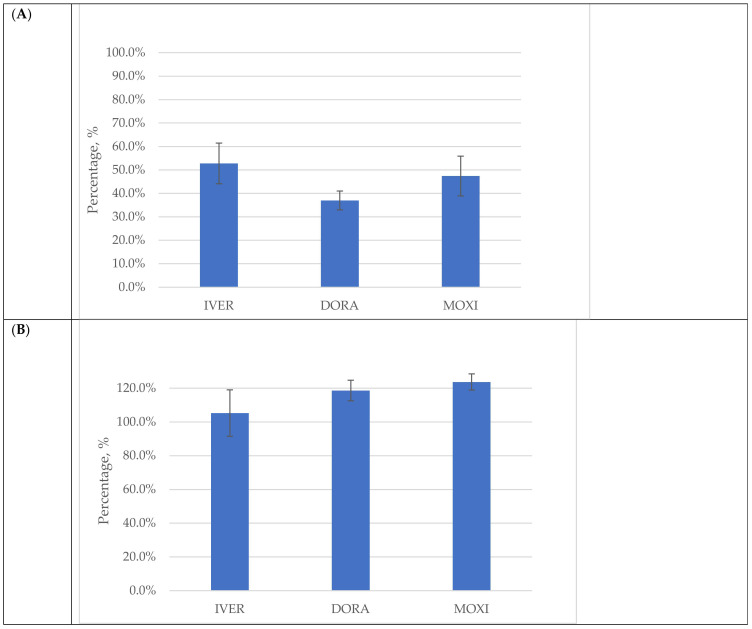
Results (*n* = 3) of the evaluation of extraction recovery (RE, panel **A**) and signal suppression or enhancement due to the matrix effect (ME, panel **B**) in bovine plasma spiked with 100 ng/mL ivermectin (IVER), doramectin (DORA), and moxidectin (MOXI) in using three different biological samples in each series (*n* = 3) after sample deproteinisation using 1% formic acid (FA) in acetonitrile (ACN) followed by OstroTM 96-well plate pass-through clean-up.

**Figure 4 molecules-27-00998-f004:**
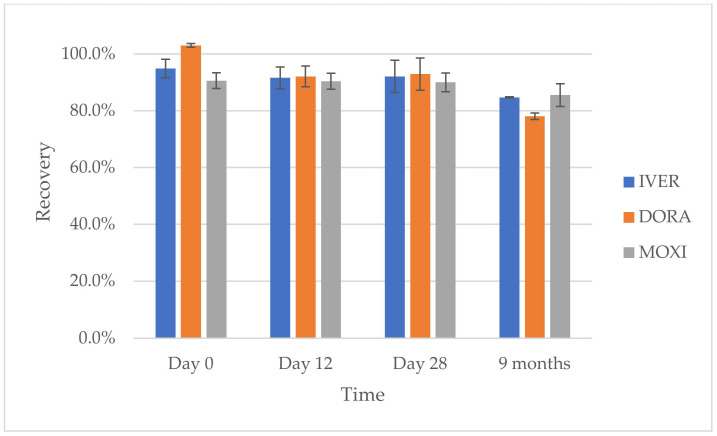
Analyte recovery of ivermectin (IVER), doramectin (DORA), and moxidectin (MOXI) in acidified bovine plasma (0.5% formic acid in plasma matrix, pH ~4) after long-term storage at ≤−15 °C.

**Figure 5 molecules-27-00998-f005:**
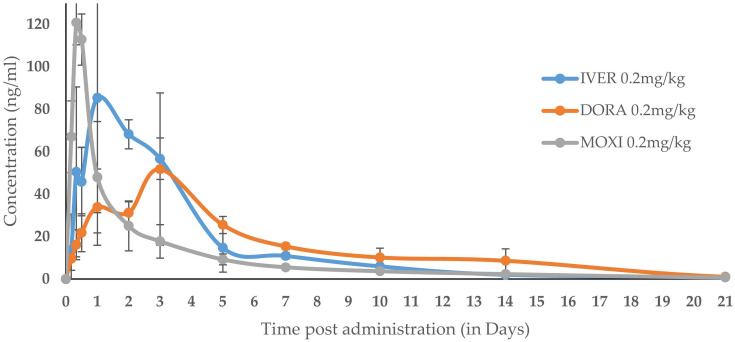
Plasma concentration versus time curves of IVER, DORA, and MOXI in bovine plasma after the subcutaneous administration of Ethiopian zebu cattle bovines (*n* = 3) with a single dose of 0.2 mg/kg BW.

**Table 1 molecules-27-00998-t001:** Analyte specific MS/MS parameters obtained in the positive electrospray ionization mode.

Analyte	Precursor Ion (*m/z*)	Product Ion (*m/z*)	Dwell Time (Second)	Cone Voltage (V)	Collusion Energy (eV)	Retention Time (min)
Ivermectin	897.50 [M + Na]^+^	329.20 ^b^	0.050	50.00	46.00	7.25
		753.40 ^a^	0.050	50.00	40.00	
Doramectin	921.70 [M + Na]^+^	353.20 ^b^	0.025	50.00	48.00	6.70
		777.40 ^a^	0.025	50.00	37.00	
Moxidectin	640.40 [M + H]^+^	123.10 ^a^	0.025	50.00	18.00	6.65
		622.20 ^b^	0.025	50.00	14.00	
Ivermectin-d2	899.50 [M + Na]^+^	183.1 ^b^	0.050	50.00	48.00	7.23
		331.3 ^a^	0.050	50.00	48.00	
Moxidectin-d3	643.50 [M + H]^+^	123.00 ^a^	0.025	50.00	20.00	6.61
		625.20 ^b^	0.025	50.00	11.00	

Note: ^a^ = product ion used for quantification, ^b^ = product ion used for confirmation.

**Table 2 molecules-27-00998-t002:** Results of the evaluation of the calibration model (correlation coe€cient (r) and goodness-of-fit coefficient (GoF), mean + standard deviation, (*n* = 3), the limit of quantification (LOQ), and limit of detection (LOD) for IVER, DORA, and MOXI in bovine plasma.

Analyte	Calibration Range (ng/mL)	r (Mean ±SD)	gof (Mean ± SD, %)	LOQ (ng/mL)	LOD (ng/mL)
IVER	1–500	0.9997 ± 0.00026	1.91 ± 0.89	1	0.02
DORA	1–500	0.9985 ± 0.00117	4.58 ± 1.97	1	0.03
MOXI	1–500	0.9990 ± 0.00045	3.86 ± 0.91	1	0.58

Note: Acceptance criteria: r ≥ 0.99 and GoF ≤ 10%.

**Table 3 molecules-27-00998-t003:** Validation results for within-day and between-day precision and accuracy of ivermectin (IVER), doramectin (DORA), moxidectin (MOXI) in bovine plasma.

Within-Run Accuracy and Precision					
Analyte	Theoretical Conc.(ng/mL)	Mean Conc. (ng/mL)	SD (ng/mL)	Precision (RSD, %)	Accuracy (%)
IVER	1	0.99	0.04	4.5	−1.2
	5	4.98	0.09	1.8	−0.4
	50	48.89	0.52	1.1	−2.2
	250	250.27	5.92	2.4	0.1
DORA	1	1.01	0.05	4.7	0.8
	5	5.09	0.33	6.5	1.8
	50	47.86	2.41	5.0	−4.3
	250	235.00	11.45	4.9	−6.0
MOXI	1	1.13	0.03	3.1	12.7
	5	5.037	0.30	6.0	0.7
	50	49.257	2.36	4.8	−1.5
	250	247.271	6.31	2.6	−1.1
** Between-Run Accuracy and Precision **					
**Analyte**	**Theoretical** **Conc. (ng/mL)**	**Mean Conc. (ng/mL)**	**SD** **(ng/mL)**	**Precision** **(RSD, %)**	**Accuracy** **(%)**
IVER	5	5.09	0.16	3.1	1.7
	50	49.17	1.13	2.3	−1.7
	250	258.42	11.56	4.5	3.4
DORA	5	5.20	0.26	5.1	4.0
	50	49.35	2.36	4.8	−1.3
	250	251.30	18.15	7.2	0.5
MOXI	5	5.15	0.26	5.0	3.0
	50	50.75	2.37	4.7	1.5
	250	248.57	20.19	8.1	−0.6

Note: SD = standard deviation; RSD = relative standard deviation (%); acceptance ranges for within-run precision: RSD < 25%, <15% and <10% for analyte concentrations ranging ≥1 to <10 ng/mL, ≥10 to <100 ng/mL and ≥ 100 ng/mL, respectively; between-run precision: RSD ≤32%, 23% and 16% for analyte concentrations ranging ≥1 to <10 ng/mL, ≥10 to <100 ng/mL and ≥100 ng/mL, respectively; acceptance ranges for accuracy: −20% to +10%, −30% to +10%, −40 to +20 for the analyte concentration of ≥100 ng/mL, ≥10 ng/mL to <100 ng/mL, ≥1 to <10 ng/mL (VICH, GL49).

**Table 4 molecules-27-00998-t004:** Evaluation of the stability of ivermectin (IVER), doramectin (DORA), and moxidectin (MOXI) in bovine plasma during three freeze/thaw cycles and in extracted samples (*n* = 3 replicates).

Freeze-Thaw Stability Samples (≤−15 °C to Room Temperature)				
	Theor. Conc. (ng/mL)	Mean Conc. ± SD (ng/mL)	RSD %	Acc %
IVER	5.00	5.18 ± 0.08	1.5	3.5
	50.00	48.26 ± 0.39	0.8	−3.5
DORA	5.00	5.24 ± 0.05	1.0	4.9
	50.00	52.37 ± 0.19	0.4	4.7
MOXI	5.00	4.95 ± 0.21	4.2	−0.9
	50.00	49.91 ± 2.82	5.6	−0.2
**Stability in Extracted Samples (for 12 Days at 2–8 °C)**				
	**Theor. Conc.** **(ng/mL)**	**Mean Conc. ± SD** **(ng/mL)**	**RSD %**	**Acc %**
IVER	5.00	5.14 ± 0.07	1.3	2.9
	50.00	48.24 ± 0.66	1.4	−3.5
DORA	5.00	5.23 ± 0.04	0.7	4.6
	50.00	50.71 ± 0.28	0.6	1.4
MOXI	5.00	5.23 ± 0.04	0.7	4.6
	50.00	49.12 ± 0.33	0.7	−1.8

Note: standard deviation, SD; Relative standard deviation, RSD; Accuracy, Acc.

## Data Availability

The data presented in this study are available on request from the corresponding author.

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
