# Peer review of "Development and Validation of a Reliable UHPLC-MS/MS Method for Simultaneous Quantification of Macrocyclic Lactones in Bovine Plasma"

_molecules, 2022, doi:10.3390/molecules27030998_

Round 1
Reviewer 1 Report
The paper presents the method for determining three macrocyclic lactones in bovine plasma. The research was appropriately designed and performed. However, I have two significant remarks. Firstly, the method is not novel. Although the sample preparation is new for this group of analytes, the authors only applied the method provided by the cartridges manufacturer. Secondly, I cannot link the need for the bioanalytical method with malaria (in the abstract, it appears to be the reason for starting the research). Treating cows with endoctocides can not be viewed as a way to control malaria. It will kill only those mosquitos in contact with the cows, and I do not think it would be the majority of the population. What correlation between PK of macrocyclic lactones in bovines and malaria do authors expect?
My technical remarks include:
Some sentences from the template are still left in the manuscript.
How were recovery and matrix effects evaluated? Was one single sample used for that purpose? If so, please repeat the experiment with different samples from different donors because it is the reproducibility of the matrix effect that is relevant for the method's performance.
The method was not validated according to CD 2002/657/EC. It is a document for residue control, not for PK purposes.
There is a 6% difference in accuracy between repeatability and reproducibility studies. It is a bit troubling. Did the authors check the results with ANOVA?
Was the type of modifier verified for MOXI? Why is there not a result for that in the Supplementary?
The stability of derivatives in FLD methods for macrocyclic lactones depends on the method (different derivatization protocols).
Reviewer 2 Report
This work described an UHPLC-MS/MS method to measure ivermectin (IVER), doramectin (DORA) and moxidectin (MOXI) in bovine plasma samples. Although the sample preparation and instrumentation are not the newest, the technical application is novel and meaningful for monitoring veterinary residues to protect food safety. I think this work can be published in this journal after moderate revision. The detailed comments are listed as follows:
- As an application method, I suggest that LOD and LOQ should be calculated based on blank real bovine sample. S/N is only for instrumental response regardless of matrix effect.
- Spiking recovery should be measured based on analyte addition before extraction. Post-extraction spiking recovery does not truly reveal the extraction efficiency, although we can obtain good data.
- More real bovine samples should be measured by this proposed method to further prove the method availability.
Reviewer 3 Report
The manuscript presented by M. Devreese and coworkers described the optimization of an UPLC-MS/MS method for fast, accurate and reliable simultaneous quantitation of the macrocyclic lactones commonly used against veterinary ectoparasites. The analytical protocol was developed according with the EMA rules, and particular attention was done to the preparation of the plasma samples, an important step to obtain reliable results by MS – based methodologies.
In my opinion the manuscript is recommended for publication after minor corrections.
Comments:
Pag. 5 section 2.14
Sentence : Optimization of the mS/MS parameters was performed by direct infusion ….. at a flow rate of 10 uL/min, in combination with the mobile phase ….
Question: how was performed an direct infusion analysis of the sample in combination with mobile phase? Probably the optimization of the MS/MS parameters was done by Flow Injection Analysis. Please confirm.
Comment
The MS/MS method used divert valve segment, this means that the analysis was performed only in a short window where the 3 MLs elute.
Can you comment why this approach is used.
Round 2
Reviewer 1 Report
I do not fully agree with the authors' statement on the novelty of the method but as the novelty is not the primary issue for the "Molecules" journal I find it acceptable. I recommend accepting the paper after just one clarification. Please state, whether there was one or more biological samples used in the validation study (especially for matrix effect). Three aliquots do not mean three different biological samples so I still cannot figure it out from the authors' explanations.
